# The Psychological Impact of Per- and Poly-Fluoroalkyl Substances (PFAS) Pollution in the Veneto Region, Italy: A Qualitative Study with Parents

**DOI:** 10.3390/ijerph192214761

**Published:** 2022-11-10

**Authors:** Marialuisa Menegatto, Sara Lezzi, Michele Musolino, Adriano Zamperini

**Affiliations:** FISPPA Department, University of Padova, Via Venezia 14, 35131 Padova, Italy

**Keywords:** chronic role strain, stress, chronical contamination exposure, family, personal violation, PFAS

## Abstract

Little is known about the psychosocial impact on people who live in polluted areas, and its consequences for the parental role have been neglected. This study addresses this gap, proposing qualitative research referring to the case of per- and poly-fluoroalkyl substances (PFAS) water pollution in the Veneto Region of Italy. The purpose of this study was to investigate the chronic exposure contamination (CEC) experience of parents. Semi-structured interviews were conducted with 32 parents living in the so-called ‘Red Area’ considered to have had maximum exposure. Grounded theory was used to analyse the data. The three themes to emerge were three phases of a dynamic process: shock around the discovery (phase 1), lifestyle change (phase 2), and living with PFAS (phase 3). The two transitions (loss of innocence and environmental adaptation) linked the phases. Our findings show that PFAS CEC experience is a process whereby parents need to move from the shock of discovery to adapting to the new situation in order to incorporate a change into their daily life, especially in regard to children. Two emerging aspects that characterised the process as a constant were uncertainty and health concerns, while role strains could be a stress source in the context of CEC. We suggest introducing the notion of chronic role strain (CRS).

## 1. Introduction

The publication of Rachel Carson’s *Silent Spring* in 1962 [1] marked a turning point in public environmental awareness of the importance of industrial pollution, especially by synthetic chemicals. This influential book exhaustively discussed the indiscriminate and extensive use of chemical pesticides and the damage they caused to the ecosystem and human health. Today, environmental pollution has become a global public health issue, and humans are increasingly exposed to a proliferation of toxic chemicals. According to the United Nations, global chemical production is predicted to double by 2030 [2], while plastic production could jump three- to fourfold by 2050, according to a World Economic Forum report of 2016 [3]. Among these toxic chemicals, perfluoroalkyl and polyfluoroalkyl substances, commonly known by the acronym PFAS, were an emerging issue in the early 2000s, although their severe toxicity was already well known [4]. In fact, researchers have long demonstrated that PFAS pollution has affected wildlife (bears, seals, birds, and other animals) worldwide [5]. From these research data, the first questions arose regarding the understanding of PFAS sources, environmental transport pathways, methods used to assess PFAS exposure, PFAS bioaccumulation in humans and wildlife, and consequences for human health [6], posing a remarkable number of challenges.

PFAS are a group of more than 4700 human-made chemicals containing an aliphatic fluorinated carbon chain of variable length (from 4 to 16), manufactured for diverse uses [7]. For many years the most used PFAS have been those with eight carbon atoms, such as perfluorooctanesulfonic acid (PFOS) and perfluorooctanoic acid (PFOA), the so-called long-chain PFAS. PFAS represent some of the most innovative material developments because of their particular chemical and physical properties. Due to these unique properties, including oil and water repellence, and temperature and chemical resistance, PFAS have been extensively used. Indeed, providing numerous benefits, they have been in common use since the late 1940s [8] for a wide variety of industrial applications and consumer products, such as disposable food packaging, cookware, outdoor gear, furniture, carpets, ski wax, personal care products, cream and cosmetics, textile treatments for stain and water repellency, medical devices, outdoor gear, paints and photography, chrome plating, pesticides, pharmaceuticals, and firefighting foams in frequent use at airports and military bases for firefighting and training activities [9,10]. It is essential to state this list of PFAS applications because it captures their diversity and suggests that they are part of our everyday life, with the high resistance to biodegradation that has earned PFAS the moniker ‘forever chemicals’ [11,12], resulting in exposure to and uptake by animals and humans. In summary, it is well recognised that, by virtue of their physico-chemical properties, their widespread use, and their extreme resistance to thermal, chemical, or biotic degradation, which causes environmental persistence, PFAS pose serious concerns to human and environmental health [13].

In fact, concerns over the potential of PFAS to adversely affect human health arise from the ease with which they are absorbed into and distributed through the body. In general, the population is exposed to PFAS pollution through food consumption, drinking water, and house dust [14,15,16]. However, drinking water has been identified as the major source of exposure in polluted communities [6,17,18,19]. In fact, the best known PFAS pollution cases involve the water: in the USA, in the Ohio Valley [20], the Cape Fear River in North Carolina [21], and Michigan State [22,23]; in Ronneby, southern Sweden [24,25]; and in Dordrecht, The Netherlands [26]. At least 90 sites across Australia are also under investigation for elevated levels of PFAS [27].

PFAS can be defined as ‘endocrine disruptors’—exogenous substances that might affect the endocrine system, compromising many of the processes of the human body [28]. Current scientific research reports that human exposure to PFAS has been associated with adverse effects on the immune [29], endocrine, metabolic, and reproductive systems (including fertility and pregnancy outcomes), and increased risk of cancer [30,31]. Epidemiological studies have reported that other health issues associated with high PFAS exposure are hypertension, higher cholesterol levels, ulcerative colitis, thyroid disease, pregnancy-induced hypertension [6,20,32,33,34,35,36], platelet alterations, and cardiovascular diseases [35,37].

In the field of industrial and occupational exposure, many studies have shown increased incidence of cancer, including bladder, kidney, prostate, and liver [38,39,40], leukaemia [39], kidney disease [31], and elevated cholesterol [41,42]. These data have been confirmed by a longitudinal study [43] on a cohort of 1002 Swedish individuals. However, there are still major gaps in translating knowledge about the health effects linked to PFAS exposure into clinical guidance.

The existing literature on environmental pollution suggests that the experience of being exposed to toxic substances can be psychologically stressing, generating a complex range of discomforts: insecurity, disruption of routine, concerns, anxiety, sense of helplessness, and anticipatory fears and disorders [44,45]. In particular, stress affects family roles. Role strain was first introduced by Goode (1960) [46] to describe difficulty in meeting the expectations of roles and balancing multiple, at times conflicting, ambiguous, or overloading roles, as well as the stress that results from the differing demands and expectations associated with the social role [47]. Role strain is usually experienced as heavy and fatiguing. In the literature on role strain, family relationships between parents and children are generally centred on general problems, such as deviant behaviour, poor academic performance, and aggression towards parental authority [48]. In this paper, we suggest the extension of the role strain concept to environmental disasters. In fact, in the case of environmental pollution, the daily routine is interrupted by a dramatic and unforeseen event for which parents are unprepared. This gives them the burden of additional educational effort with no certainty of being able to meet the expectations and responsibilities of their role.

In the Love Canal environmental pollution affair, Stone and Levine [49] found a high level of strain in families. Analysing the Legler affair, a case of water pollution caused by a municipal landfill, Edelstein [50] highlighted the impact on family dynamics, both among couples (conflicts and separations) and in the parental role. For example, parents’ concerns about child safety lessened the quality of family life, especially when coupled with the difficulties of teaching new behaviours, such as being afraid of tap water [51]. As regards PFAS pollution, specifically, role strains on parents have additional aspects. In fact, as evidenced by research [36,52], PFAS accumulated in the mother’s body can be transmitted to the unborn child through the placenta and during lactation. This intergenerational transmission from mother to child biologically intoxicates the unborn child and could psychologically undermine the mother’s childcare and protection role. Despite a recent increase in research focusing on the effects of PFAS on physical health [53], there is as yet, unfortunately, little research analysing the psychological impact [54], and its consequences on the parental role remain neglected. The purpose of this article is to contribute to filling this research gap. The case under examination concerns an episode of PFAS water pollution in some areas of the Veneto Region (Northern Italy) associated with the activity of an industrial plant located in the area where, for decades, PFAS were produced.

## 2. PFAS Pollution in the Veneto Region, Italy

In late spring 2013, a vast area of the Veneto Region of north-eastern Italy was found to be polluted by PFAS. This was revealed by a study begun some years before, in 2011, when the Italian Ministry for the Environment, Land, and Sea commissioned the Institute of Water Research of the National Research Centre (IRSA–CNR) to evaluate PFAS pollution in major Italian river basins [55]. This study was triggered by the European PERFORCE project, an investigation launched in 2006 with the aim of assessing the exposure of perfluorinated substances in the European environment, including the waters of European rivers. In Italy, the Po River, located to the north of the country, showed the highest concentrations of PFOA [56], and a few years later it was the turn of the Brenta River basin in the Veneto Region to show certain ‘hot spots’ where PFAS were found in both surface- and groundwater [57].

After this pollution was discovered, the Agency for Environmental Prevention and Protection of the Veneto Region (ARPAV), together with other regional authorities, took a series of measures to determine its extent and level and to limit human exposure to PFAS [58] through an environmental monitoring plan.

The results showed high levels of pollution in three provinces: Vicenza, Padova, and Verona (see Figure 1); the pollution involved groundwater, surface freshwater, and drinking water, and hundreds of thousands of people were at risk. Initially, an area of maximum exposure of about 200 km^2^ was identified [59] and named the ‘Red Area’. It was composed of 21 municipalities with 126,000 inhabitants. In 2018, nine municipalities were added. At present, the ‘Red Area’ is 595 km^2^ wide and has a total population of approximately 140,000 people [60]; however, the extent of the territory involved will increase due to water mobility. For decades, residents and workers of these communities have been exposed to PFAS through the use of contaminated ground (bore) water and town water, as well as from local produce, among other exposure pathways. This technological environmental disaster is considered the third-largest pollution event worldwide due to the extent of the water pollution involved [58].

The source of the pollution was mainly associated with the activity of an industrial plant located in Trissino, in the province of Vicenza. The Miteni Group (formerly called Rimar), a reference company in fluorine chemistry which has produced PFAS since 1968, was identified as responsible for the disaster [57]. For many years, the factory has continuously produced PFAS-releasing waste and wastewater without any environmental protection or security and safety measures due to a lack of specific regulations.

In July 2013, activated carbon filters were installed by water service companies in the plants treating the public drinking water distribution system. At the same time, private wells were mapped, and citizens were obliged to assess PFAS levels in them and to avoid using water where pollution was found. In 2016, a biomonitoring study was conducted on a randomly selected group of citizens living in the polluted area and on non-exposed citizens in order to assess PFAS exposure. The study was designed and conducted by the Italian National Institute for Health (Istituto Superiore di Sanità, ISS) in collaboration with the Region and with the support of the Prevention Departments of Local Health Authorities (Aziende Unità Locali Socio-Sanitarie, AULSSs) with specifically appointed personnel, including Environmental Health Officers and Health Visitors. The results showed that the median value for the PFOA concentration was about eight times higher in locals than in non-exposed people; it tended to be higher in men than in women, and the consumption of tap water was positively correlated with its concentration [61]. The same study observed that PFAS concentrations in the serum of exposed farmers and breeders were higher than in those of unexposed people [62]. Girardi and Merler [63,64], in an observational mortality study involving male workers at the Miteni plant, found that mortality from liver cancer and lymphatic and haematopoietic tissue (LHT) malignant neoplasm was more than double, considering regional rates as reference.

**Figure 1 ijerph-19-14761-f001:**
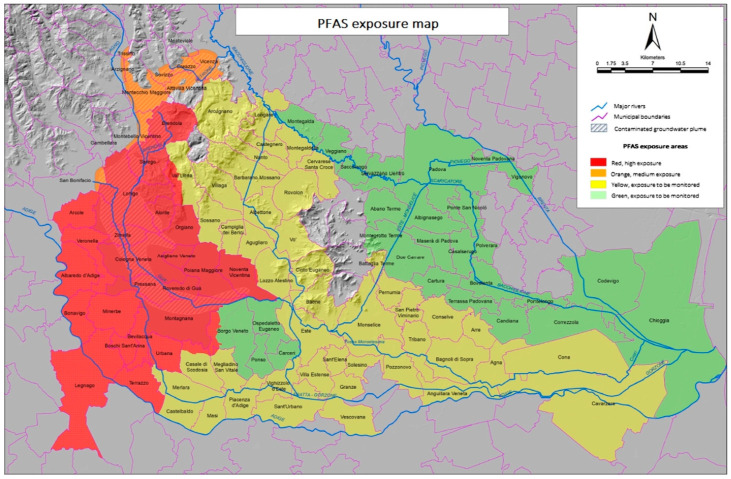
PFAS exposure map indicates the different colored areas and the Municipalities involved [62].

Since then, a health surveillance program for residents of the ‘Red Area’ has been implemented in response to public and private concerns about the health impact of exposure to PFAS pollution. The aim is to use epidemiological evidence to aid in the prevention, early diagnosis, and treatment of some of the chronic disorders. The surveillance program is ongoing.

In 2021, the trial of the managers of the Miteni factory began. They were charged with causing environmental disaster, and over 100 victims of PFAS pollution participated, demanding justice and compensation for damages (for the timeline, see Table 1).

## 3. Materials and Method

The present study is the first phase of a mixed-methods community health resilience (CHR) research project, currently in progress, based at the University of Padova (Italy) (FISPPA Department). It is designed to investigate psychosocial risk factors, psychological discomfort, community conflicts, and resilience strategies among the population of those areas of the Veneto Region (Italy) that have been exposed to PFAS pollution. The study included numerous visits to communities polluted by PFAS, during which we conversed with formal and informal leaders and engaged in many forms of observation. In combination with the existing literature on environmental pollution [50,54] and an analysis of government documents [65,66,67], these visits to communities informed the development of the topics covered in the interviews. In-depth interviews were preferred because they are more suitable to capture the concerns, opinions, perceptions, emotions, and experiences of locals, providing an opportunity to explore topics that inform the development of further studies using quantitative methods [68]. Topics covered in these interviews included the discovery of PFAS pollution, experience of being exposed to pollution, loss of resources, family life, changes in lifestyle, search for information on the harm caused by PFAS, and efforts to obtain justice and compensation for damages.

### 3.1. Recruitment and Data Collection

The recruitment of participants in qualitative research with in-depth interviews is a complex process which basically consists of creating a connection with people who have relevant experience and agree to share it [69]. In our study, we involved a grassroots movement, ‘MammeNoPfas’ (MothersNoPfas), an association committed to denouncing PFAS contamination in the Veneto Region, as a mediator. This mediator used its position and formal and informal relationships to facilitate contact between us, the researchers, and potential participants. After identifying a mediator, we planned the recruitment strategies [70]. Seeking to create a connection between researchers and participants, we first implemented a communication strategy to let potential participants know about the study and how they could become involved. We also tried to demonstrate the social value of the research, offering all the scientific and ethical information at our disposal to give credibility to the study and establish a relationship of trust. Finally, the researchers made themselves available to carry out the interviews on any day of the week and in any time slot, ensuring maximum flexibility for the participants.

General inclusion criteria for the study were to be resident in the ‘Red Area’, considered the area of maximum exposure; to have resided there since before 2013, the year the pollution was discovered; and to be a parent. Residents who met the study criteria were selected from January to March 2020.

Thirty-two community members (*n* = 25 female and *n* = 7 male) of the polluted area of the Veneto Region responded to our request and were admitted based on the study criteria. They came from different municipalities of the ‘Red Area’. Due to the explosion in Italy of the COVID-19 pandemic in February 2020 and in accordance with the consequent health emergency and lockdown measures adopted, it was not possible to conduct in-person interviews. As a result, we reconsidered our approach. As advances in communication technologies offer new opportunities for conducting qualitative research [71,72,73] and online methods can replicate traditional methods, including in-person interviews [74], we used a videoconferencing platform to collect interview data.

Two researchers (Michele Musolino and A.Z.) interviewed the participants. All interviews were between 60 and 90 min long and were audio- and video-recorded and transcribed verbatim; participants reaffirmed consent verbally prior to the online interviews. Although some participants in the study experienced some degree of difficulty in joining the session (due to initial technical difficulties and low Internet bandwidth), all described their online interview experience as satisfactory and convenient, particularly in terms of time effectiveness, given their location and busy work schedules. As researchers, this technology gave us a wider geographical reach by extending our recruitment strategy to all municipalities in the ‘Red Area’. All interviews took place between April and July 2020. Using the data saturation strategy suggested by Charmaz (2014) [75], we considered stopping at 32 participants because no new insights or themes emerged thereafter in the data gathered [76]. Our sample size for saturation also conforms to the research parameters obtained through a systematic review by Hennink and Kaiser [77].

We obtained written informed consent from all participants prior to the online interviews. This study followed the American Psychological Association Ethical Principles of Psychologists and Code of Conduct and the principles of the Declaration of Helsinki and was approved by the Ethics Committee of the Università degli Studi di Padova (Protocol Number 1D4BA484CC28FCDA6984C4F21E59DEA6).

### 3.2. Data Analysis

The data were analysed using thematic analysis [78], a qualitative method used for analysing data by identifying patterns and organising them into themes. The analysis process was in line with the phases of thematic analysis identified by Braun and Clarke (2006) [79]. The software package Atlas.ti was used to facilitate qualitative data analysis. Each transcript text was initially read and reread in an active manner to allow the researchers to become familiar with the data, after which initial codes were created through the selection of text parts, named quotations; thereafter, codes were grouped into categories and categories into themes. The coding scheme was developed both deductively and inductively to allow for the inclusion of several patterns suggested by the literature and to consider specific elements present in the data. A thematic structure was developed with themes and subthemes. Two independent coders and a third external coder analysed the data.

Themes that were endorsed totally by participants were considered prominent themes and subthemes that were endorsed by half or more of the participants were considered prominent subthemes; thus, these themes and subthemes are described in depth throughout the results section. Descriptors were systematically used to denote the number of community members who participated in the study and endorsed each theme and subtheme, where ‘no’ indicated 0% of the group, ‘few’ indicated 1–25%, ‘some’ indicated 25–50%, ‘many’ indicated 50–75%, ‘most’ indicated 75–99%, and ‘all’ indicated 100%. Such descriptors are conventionally used in thematic analysis to represent the prevalence of themes and subthemes in the data [79].

## 4. Results

### 4.1. Participant Demographics

Thirty-two community members participated in the study, of whom *n* = 25 were female (mothers) and *n* = 7 were male (fathers). They ranged in age from 39 to 65 (Mean = 52.63; SD = 7.53); *n* = 27 participants were married, *n* = 3 were living with a common-law partner, and *n* = 2 were separated/divorced. Twenty-three had received a middle-level education (71.8%). All were parents of at least one child and a maximum of five. As concerns the children of participants, *n* = 31 were under 18 years old, ranging in age from 1 to 18 (Mean = 12.52; SD = 4.37), and *n* = 34 were over 18, ranging in age from 19 to 43 (Mean = 26.35; SD = 5.74). All participants came from municipalities located in the ‘Red Area’, where they had resided for an average of 35.86 years, ranging from 10 to 62 years. All owned the houses in which they lived, most of which had gardens, vegetable gardens, and private wells, and had, in many cases, been inherited from parents and were therefore the residences where participants had spent their childhood, having been rebuilt over time (see Table 2).

### 4.2. Themes, Subthemes, and Transitions

All findings were summarised into 3 themes and 10 subthemes. The themes and subthemes are represented as three different phases—parts of a dynamic process determined by the interaction of parents with the traumatic event due to the PFAS pollution. In this psychological impact process, two subthemes (uncertainty and health concerns) emerged as constants, because they are present in all textual data shown in the three phases. These three phases denoted how parents dealt with the PFAS event over time, while the two transitions (loss of innocence and environmental adaptation) indicated the parents’ need to adapt to the new situation, or circumstances, in order to incorporate a change into their daily lives (see Table 3).

#### 4.2.1. Phase 1: Shock around the Discovery

This theme emerged when the parents involved in the study described the moment they discovered they were living in PFAS-polluted land and, a few years later, of being personally polluted. In general, parents learned about the pollution after the results of the early studies, when the first news began to circulate in the community through local media articles or was passed by word of mouth among local citizens in 2013–2014. This circulation of knowledge was the cause of serious social alarm and deep concern to residents; as one mother said, “we were very worried and lived in apprehension”. In 2017, it was discovered that all participants were part of a polluted community when their children were recruited for the first biomonitoring study and the results of the blood testing undeniably determined high levels of PFAS. All parents reported an initial reaction of shock and devastation following the awareness that the pollution had affected the bodies of their children. They declared that the discovery was surprisingly sudden, powerful, and a difficult situation to deal with. They reported being confused after hearing the bad news and experiencing a sense of surrealism and disbelief; for example, one parent said that “it was like being in a film”, while another stated, “I was shocked… It took me about an hour to come back to myself, I lost my mind”. Another parent, a mother of two adopted daughters who were tested, said, “when I saw their results, I was hurt… I had told myself that they were lucky as they hadn’t lived here for many years before being adopted… instead, they had high levels, I felt truly lost”. Most mothers argued that it was a shock to see the analysis written on the report with the name of their own child, as conveyed in the following quote: “discovering something unknown, which has never been seen before, was indescribable; you cannot express it in words. It is a profound and incomparable shock”.

##### Unanticipated Event

This subtheme refers to one characteristic of the event that attacked their global belief system. The parents said that their sense of shock was due to the fact that the event was unanticipated. For decades prior to 2013, inhabitants had known nothing of the severity of the pollution, so they never imagined that, on one day in their life, a dangerous substance would taint the tap water. As one mother reported, “my municipality being a tanning district, it has always been famous for pollution. There were occasional accidents, such as a stream that became discoloured, after which everything seemed to return to normal, local institutions giving us reassurances”.

Concerning the Miteni plant, only one parent said they remembered this name (in reference to Rimar becoming Miteni); however, they did not know what it produced exactly, merely considering it one of many local companies in the tanning district. This parent is a father who said, “I only remember its name. When I was young, my grandparents and my mother told me about an incident that happened during the 1970s at a local plant. In practical terms, the tap water was brought in by tankers. This incident was attributed to the Rimar”. The other parents knew nothing of the existence of the Miteni plant before the discovery of the pollution. Further, all reported that this plant is located in an area completely hidden by trees; in fact, one mother said, “I didn’t ever see it! Yet, I used to pass by it! It is completely surrounded by large trees”.

Regarding the scale of the event, no one could have imagined that the polluted area was so big, the second-largest in Europe, the size of Lake Garda. As one mother said, “I never expected this, we have a polluted aquifer as big as Lake Garda, completely compromised. It’s unacceptable, because we’ll never use this water”. Considering the relationship with institutions, the parents would never have expected that they would underestimate the situation and fail to take safety measures; in addition, they could not believe that the control and security authorities had not previously been aware of the pollution. In fact, all parents declared that “they had trusted the institutions, and this massive problem wasn’t supposed to happen”. Most parents said that, beyond the shock of the change to their own and their children’s condition, there was also the shock of an unexpected and incredible situation. They had entered a world which would, to that date, have been completely unthinkable: as one father said, “unimaginable, we didn’t imagine a thing like this could happen”. This triggered the need to make new sense of the experience.

##### Ignorance and Information Needs

When the participants learned they had been exposed to toxic chemicals for decades and their children had high levels of PFAS, they wondered what it meant for their health. At the beginning of the period of social alarm, little information was immediately available; later, with the biomonitoring study, they felt that they had not been offered either sufficient health information or professional support to cope when they found out the results of the blood tests or the necessary information to protect themselves, their family, and their children. In both cases, most parents stressed that information about PFAS had been absent, incomplete, and insufficient. One parent said that “at a personal level of knowledge, at the beginning of the discovery, we didn’t even know what PFAS were”. Another mother argued that “at a community and institutional level, we should also be provided with guidance on actions to take”. With regards to the biomonitoring test, many parents affirmed that “in the absence of information about health effects or clear guidelines regarding how people can protect themselves, testing could do more harm than good because test results could merely raise concerns without any helpful support offered”. All parents were motivated to access as much information as possible, even if they met considerable difficulty and obstruction doing so. Most participants reported having sought information from the first accessible resource, namely the Internet. One participant mother declared, “I immediately looked for more information on the Internet, helped by my son... Not to go adrift… I needed to clear my head”. In addition, participants described a variety of human sources consulted to learn about pollution. Many parents used personal contacts within the community, asking for information from neighbours, acquaintances, community leaders, family doctors, and work colleagues: “I met the priest and I asked him, I met my doctor and I asked him… but nobody knew”. All parents experienced that, at community level, information was confused, negligible, and full of contradictions.

##### Personal Violation

From the results descripted above, one consequence of the discovery of pollution can be identified as emotional impact. This experience is akin to a traumatic event, when this causes a strong emotional experience. Feelings aroused from the discovery of the PFAS pollution involved several negative emotions. When parents were asked to define the emotional consequences of shock, they reported feeling numb, dazed, panicked, afraid, and worried. Words such as sad, devastating, heartbreaking, and sense of alarm were also used to illustrate the emotional impact of the discovery of PFAS pollution. All parents declared that the discovery was a dramatic experience, especially given their position as the parents of the children tested. One mother remembered the shock when “the results of my sixteen-year-old daughter arrived by an email, she had eleven times the tolerable amount of PFAS in her blood”. However, anger and worry were the main manifestations of the feeling of personal violation. One mother said, “I felt that my body, my privacy, my family had been stolen; it is as if thieves came into my home from the tap”. Another mother referred to the results of the blood testing in these terms: “I couldn’t accept that these substances had come into my blood, my body, through drinking water and food… I never really thought it would happen to me”. Participants captured and emphasised that they were unsafe at home because home had become a place where they felt threatened, rather than a protective place. One father said, “we want a home free from these dangerous chemicals; they are everywhere”, while a mother reported, “I consume water inside my home, a place where I should be safe… I don’t feel protected”. Fear was the most commonly elicited emotion. This illustrated the threat of being a victim (direct or indirect) of a potential disease or the worsening of pathologies already in progress. Mothers placed more emphasis on these emotions than fathers, showing greater emotional expressivity, especially stressing their feelings of fear, worry, and anger.

##### First Transition: Loss of Innocence

The ‘loss of innocence’ represented the transition from shock to awareness of the pollution.

Specifically, the discovery started when parents perceived a threat, through some external information, based in the polluted land. Although the discovery initially referred to something completely unknown, the initial alarm was linked to the extent of the pollution due to the industrial exploitation perpetrated by the polluting company, whose existence had previously been unknown to all interviewees. One father said, “I realised the crime and madness of the industrial system; we have lost groundwater as large as Lake Garda”. Through the health surveillance program and biomonitoring test, pollution became something that affected the parents and their children. Human biomonitoring plays a fundamental role in assessing the degree of pollution; it removes all doubt and hope, forcing parents to recognise that PFAS contamination affects them directly. One mother underlined that “now we are all poisoned”.

#### 4.2.2. Phase 2: Lifestyle Change

Since it was no longer possible to ignore the situation, parents described a process of protecting their children and family and coping with the PFAS pollution. All participants inquired about the potential health implications of the pollution and, accordingly, reported that their lifestyle necessarily changed: they reviewed their life, re-evaluating and changing their priorities to seek safety and security. Due to the PFAS pollution, there was a sudden and radical change in the lives and habits of the parents, obliging them to make healthier choices and adopt a healthier lifestyle.

In general, there were two aspects to the change: the first, controlling the risk of pollution consisted of reducing, even eliminating, the chemicals in their own home; the second consisted of adopting prevention measures in order to look after their health and avoid chronic pathologies or other adverse health consequences. Maintaining a healthy nutrition status was crucial; it was therefore unsurprising that all parents referred to controlling the primary source of pollution, the drinking water, alongside food. Referring to dietary habits, maintaining a balanced diet, and paying attention to food frequency, such as weekly meat consumption, or other food likely to have been in contact with PFAS-polluted packaging, such as pizza boxes, emerged as a new lifestyle, especially in regard to children. Indeed, all parents paid extra attention to the lifestyle of their children, such as ensuring they ate a healthy and balanced diet, including their education.

This new role, of being careful about what their children eat and drink and educating them to adopt a healthy diet, was considered worrisome and laborious by many mothers. For example, one mother said, “every morning, it is not easy to tell my son to take bottled water, to be careful about the food at school; I haven’t got into the habit of doing it yet!”. Concerning lifestyle habits, it changed the way parents bought their groceries. As one mother stated, “before, I used to buy food just about anywhere, for convenience. Now, I’m very careful about everyday choices, above all for food; for example I buy it in *Gruppi di Acquisto Solidale* (GAS; ethical purchasing collectives). Moreover, adopting a vegan diet was an important change for one mother and her family: “I adopted a vegan diet for me and my husband. I told my daughter what had happened and the reason we were eating vegan food. I also buy vegan biscuits for the dog”.

Practising physical activities, or doing so more frequently, also emerged as playing an important preventive role. For instance, many parents declared that they went walking more in the open air of the mountains. Many interviewees described how, over time, they formed a new lifestyle routine, especially in regard to their children. This ‘educational work’ not only increased after the discovery of the PFAS pollution, but changed quickly, with little notice, and with many difficulties: as one mother reported, “it was very difficult for me! At first my children refused these new styles; they didn’t want to hear! They didn’t understand the danger, not even my concerns. They came to understand it gradually”.

##### Stressful Role Demands

All 32 interviewees expressed their discomfort dealing with the PFAS pollution and its implications for their own family roles and health obligations. Pollution was the starting point in the adoption of new educational roles by parents, which had to be seen as more important than previously. As one father said, “before the pollution we lived just a normal routine. Now it is as if a chain had broken into various parts. I must think and reason about every link in the chain! I must plan and re-evaluate everything, especially for our child. This is a source of great discomfort in our lives”. Indeed, parents had to re-teach relationships with water, in particular, which was commonly reported by many parents as “really hard work”. A prevalent manifestation of difficult role demands among parents was their concern about their children’s health and their ability to educate them to apply new safety rules, why the new rules were necessary, and, above all, what the new dangers were, such as being afraid of the water. One mother explained it well: “I taught my children that water is a danger… to be careful when cooking and washing, also when we give water to our dog, and then, with time, I explained something to them about the pollution caused by the Miteni Company”. These new roles included teaching the children to be careful what water they drink, to take bottled water everywhere, and not to drink tap water in schools, gyms, and football clubs.

Even if the parents both reported being involved in and sharing this educational role, many mothers faced increasing stress when taking up the role of ‘supervisor’ and ensuring the children observed the new rules. This parental burden was more intense at the beginning, when parents had to step into the new role of coaching their children, helping them to recognise danger, and remaining alert and watchful. One mother stated that, “since we had to stock up on water, I had to teach my son how to do it. But he often forgot so I got nervous all the time, it was hard for me to remind him every time and to make sure he did it”. According to another, “above all, at first, it was difficult, we always had to be careful, always telling him what he can and can’t do or interrogating him about what he had eaten at school”. Interestingly, another mother, by profession a primary school teacher, declared that she had found it hard to decide what to convey to her daughter about safety without scaring her: “finding the balance between fear that protects and fear that inhibits can be very difficult”.

##### Forced Protective Behaviours

Faced with the reality of the pollution, parents were forced to take action immediately, before illnesses occurred. The discovery of PFAS pollution motivated the parents involved in this study to adopt protective behaviours and provided a framework for personal safety, allowing families to feel safe in their homes. For most parents, these behaviours initially consisted of no longer drinking water from the tap and using alternative sources: most declared they had started purchasing bottled water, and one chose to obtain PFAS-free water from mountain sources far from the polluted land. The water from alternative sources was used by participants to cook all types of foods, for personal hygiene, and for pets. As one participating mother reported, “our water use was completely changed… We exclusively used tap water for washing dishes or doing laundry… We no longer use it to cook food, even to brush our teeth… We don’t even give it to our dog”. A father said, “we started immediately to cook food with bottled water, everything! Such as pasta, soup, herbal tea, coffee; we are still doing it now”. The water from private wells was no longer used to provide small summer pools, irrigate vegetable gardens, and raise livestock, leading these activities to come to an end. Consequently, many participants who had vegetable gardens have let them go completely.

Once they had completed controls on the primary polluted resource, domestic water, all parents focused their efforts on removing all of their PFAS cookware from their house. They also removed from the pantry any items that could contain unsafe chemicals in the food packaging, such as microwave popcorn bags and plastic containers. In addition, checking the origins of local food, especially fish, eggs, and vegetables, and no longer consuming certain traditional local products, were practices that have continued into the post-discovery period to deal with the pollution. All parents who endorsed this subtheme supported measures to reduce pollution, being particularly careful about the food supply chain to eliminate potentially dangerous food produced in polluted areas. As one father stated, “at the beginning, we eliminated local market foods, or in any case, any food that we knew came from the polluted area”. However, this implied a cognitive fatigue described well by one mother: “I have to be very careful about grocery, I take four times as long, and this leaves me exhausted”. Additionally, an economic burden emerged; in fact, a few participants installed private reverse-osmosis filters, reducing PFAS levels in tap water, and purchased new PFAS-free products for themselves and own children, including upholstery and carpets, water-resistant clothing, personal care products such as dental floss, and cleaning products. Interestingly, one mother started making detergents at home.

##### Second Transition: Environmental Adaptation

Parents’ stressful role demands, together with their commitment to changing lifestyle, take us to the second transition, namely environmental adaptation. Two key elements in the adaptive experience of the parents were identified. The first was a private proactive behaviour consisting of an independent search for information (including the Internet) or authoritative advice (scientific literature), by which the parents became aware of the gravity of the pollution. A second crucial element was identified as self-management of chronic pollution, a process through which individuals actively coped with their chronic disease in the context of their daily lives. In fact, as the parents could not eliminate the PFAS from their environment, their efforts were concentrated on containing and mitigating damages. All interviewed parents were aware that they could only deal with this situation reactively. Therefore, for all parents, adaption and mitigation actions and measures became increasingly necessary for protection and safety. As one father said, “we adapted. It’s like for those who live in a seismic zone and have to adapt”. However, one mother described well the process from the first transition to the second one: “at the beginning, when there were no water filters, using bottled water to boil pasta was troublesome, very difficult for me to manage, very stressful, on top of the economic impact on the family. Then we adapted, and the change has become a habit”.

#### 4.2.3. Phase 3: Living with PFAS

This theme reconstructs the time process, establishing connections from the past, the discovery of PFAS pollution, and the loss of innocence, with the future; thus, it represents a connection with the passage of time that characterises the transition through chronic pollution. When interviewed parents were asked to describe their experience of living with PFAS pollution, they generally answered that “we find hard to leave our roots”, so they decided to continue living in the area and adapted. One of the main reasons for this is because they have historically, generation after generation, lived in that area. In many cases, their home was inherited from parents or grandparents, leading to a particular place attachment. As one mother reported, “this is the house of my grandparent, that he bought thanks to hard work in the mines in America. He was an emigrant post-war. So, for me, this home has an enormous emotional value”. Another reason is family commitment. All the parents had children who had friends and schoolmates; therefore, they felt a commitment to staying there and thought it would be unfair and disruptive to move away. Additionally, most would have found it hard to have to start over, without the guarantee of a new job. This was well expressed by one mother: “Now, we have adapted, but at first, we often thought maybe it was better to move out. But where would we go? Then, I absolutely love my job, I’m an educator in a therapeutic community for eating disorders, and it would have been difficult to find an alternative. Lastly, my daughter was opposed to the idea because she wanted to be close to her grandparents, friends, schoolmates”. Another significant reason for not relocating was that the parents did not have the funds to do so, and there was no insurance compensation, as is the case in an earthquake or fire; therefore, it was more feasible to deal with the PFAS pollution than move away from their homes. This is well reflected in the words of one father: “I had just invested a lot of money in renovating my parents’ house, I have a mortgage to pay, we have to stay there, even if I feel like I’m just stuck”. In turn, works on the local aqueduct to have PFAS-free water and a surveillance program put in place by the Veneto Region to lower the risks have made them feel prepared to deal with the effects of the damage caused by the pollution.

##### Invisible Enemy

Many interviewees commonly described critical issues related to the experience of PFAS pollution using the metaphor of ‘the enemy’. Phrases commonly cited were “water became an enemy”, “a polluted environment can be hostile, it seems to wear a mask”, “an enemy came from drinking water”, “an invisible enemy”, and “there seems to be an enemy in everything”. In fact, the parents were dealing with a great unknown: they could not verify the presence of PFAS, because PFAS are invisible and odourless; how could they know if their health was affected? In other words, whereas in other types of pollution one can see or feel certain effects, understanding the danger of water is difficult. Lastly, the PFAS in one’s own body work on a pattern of guesses, and consequently they can blow up at any time. As one mother said, “it’s like inside your blood there is a bomb that can explode at any time. We don’t know if it will happen, when it will happen and how much”, and only then does the invisible become visible in the shape of an illness. Moreover, to face chronic pollution takes so much time and energy that it feels as if one’s life is on hold.

It is like being trapped in a war against something that is everywhere, even inside you. Every day consists merely of a battle in which people are protecting themselves and their children from an invisible enemy with no real guarantee about what will work and what will not. As one father said, “sometimes, I think we are in a long-term war. I think of it as an undetonated bomb, which only causes harm when it actually explodes”.

##### Parental Guilt

Most of the participating mothers who endorsed this subtheme reported feeling guilty about the transmission of PFAS substances to their child through breastfeeding, pregnancy, or a diet they believed was healthy and chemical-free. One mother who felt persistent guilt felt as if she were the only one responsible for having polluted her son: “I feel very guilty, maybe I have delegated to others something that I should have done… as parents, we had to be very careful”. Many, reflecting on their pregnancy and post-partum period, remembered these as troubled and difficult to handle: “I was the first one to pollute my daughters with PFAS through pregnancy and then by breastfeeding; this is a very hard feeling of responsibility to bear. As a mother, it is very difficult to endure. We did not commit this crime, but at the end, we feel guilty”. Another said: “I breastfed my daughter until she was three, thinking I was doing my duty as a mother”.

Then, when it came to supervising their children in the area of a healthy diet, mothers said that if they had known about the pollution, they would have avoided polluted sources such as drinking water or local food. As one mother reported, “I thought I was feeding my children the best vegetables, the best fruit, safe food, and they were delicious; however, I was poisoning my children and my husband”. Additionally, two mothers of adopted daughters tested, with high PFAS levels in their blood, said they felt guilty for not having given them a better life. For the interviewed fathers, guilt arose when they became aware that, as parents, they had failed to be the best they could have been for their children. As one father said, “as a parent, you try your best. We were careful they didn’t smoke, didn’t drink alcohol, that they were wearing safety belts while driving. We took various precautions for their safety. Instead, the danger has come from the water and the food that we thought were healthy, if we had known it before…”

##### Social Support

For most participants, social support played a central role in the beginning of their engagement to deal with PFAS pollution. They stated that social support was found from other community members involved in the pollution in the form of looking for and sharing information and caring for each other, which was especially helpful in finding quick solutions. Community meetings were pointed to as an important source of support by many parents. In fact, most parents declared that they had met people at initial and ongoing initiatives “with whom one could feel comfortable and had a common goal”, as one father said. At the beginning, the first initiatives were organised as informal meetings at the community level (“there was a mum who was offering her basement room for the evening meeting. At each meeting she prepared us local pancakes, so we laughed, talked, joked, even if we were desperate, and even when the meeting ended with nothing”, as one mother reported) or in previously formed groups, such as the local GAS. Additionally, and above all for many of the interviewed mothers, emotional support, especially during acute moments, was found in regular exchanges with other mothers: “there, you could also talk about things that were bothering you at the moment or just vent. Knowing that if there was a problem we would meet up shortly. That really helped me a lot”. In these meetings, the parents called experts to explain what PFAS were and their consequences for health, as well as to share concerns and, finally, to expand their knowledge of the phenomenon.

##### Civic Engagement

After the embryonic local meetings, the parents moved from one municipality to the next, involving other community members and relatives. Gradually, grassroots groups took shape. An intensive network was constructed throughout the polluted area joining diverse local communities and, at the same time, extended to other parts of the polluted area of the Veneto Region. For many parents, learning was initially improvised, after which it became part of a continuous process in their life. At this stage, most parents involved the community in the role of informal education about the PFAS pollution. For example, many parents started informing those of their children’s schoolmates. As one mother stated, “my need was to gradually inform families, neighbours, acquaintances, schools, because what had been happening was serious”. Some started to write a collective complaint from residents, collect signatures, and raise people’s awareness of the issue. A mother said, “we went to the square every Friday of the month to collect signatures. This was the beginning of informing my fellow citizens what was happening”. In this way, the parents started forming groups on social networks such as Facebook and Telegram in order to share information quickly and effectively. They also created courses and educational materials in schools.

#### 4.2.4. Uncertainty and Health Concerns

Two common subthemes emerge from the data and are present in each of the three phases of the psychological impact due to PFAS: uncertainty and health concerns. Parents explicitly expressed concerns about the health of their children. For example, one mother asked whether exposure to pollution could cause future diseases: “during pregnancy I drank water polluted by PFAS. What will be the consequences for my daughter? Who should I ask? No one can give me an answer”. A father pointed out similar concerns: “it is really worrying that no one has been able to tell us how things really went, how much pollution my children have suffered, what damage I might have caused to my children by giving them tap water to drink, and what risks I have to face”.

Once the awareness of living in a polluted environment had matured, the concerns regarded behaviours required to limit exposure and reduce damage: “I made the whole family change its diet, I strove to teach my children to be afraid of the water they drink, I even forbade my daughter to go swimming in the municipal pool, but will my children really understand? Will they follow my instructions? And will all these efforts have any effect?”. When parents had entered the phase of ‘life with PFAS’, concerns continued to characterise their daily life. “Who can assure us that the filters at the aqueduct retain the PFAS?” one father asked, and a mother reiterated that “we live everyday with danger, inside our bodies and inside the bodies of our children, scattered everywhere around us: in the soil, in our wells, in our gardens, and in the food. And we don’t really know how to neutralise it and, most of all, if we will ever manage to”. Persistent worries are always associated with a pervasive sense of uncertainty.

## 5. Discussion

The psychological impact of PFAS pollution in the Veneto Region presents aspects that are traceable in the experiences of other communities damaged by pollution by toxic chemicals [51]. All the interviewees reported stress levels and were particularly anguished about the implications for their lives and those of their relatives. Despite the installation of active carbon filters to prevent the pollution of drinking water from aqueducts, the psychological condition of the interviewees is aligned with Couch’s (1996) [80] description of a community affected by a disaster, trapped in the warning, threat, and impact stages, unable to foresee a recovery stage [81]. If, in the long term, two solutions are offered to a seriously polluted community, namely some form of clean-up of the polluted territory or community relocation [80], polluted communities in the Veneto Region currently have to live with PFAS. In fact, the extent of the pollution has not yet been fully clarified, and the large affected area is densely populated and rich in productive activities; this makes extreme measures impracticable, even if some interviewees expressed a desire to move somewhere else.

Moreover, despite the general expectation, no clean-up has been carried out yet; it is difficult to implement because the pollution has affected a large aquifer that moves subterraneously, transporting PFAS to areas at a considerable distance from the polluting industrial site. Since the environmental damage (to the territory) and the biological damage (to people) have already occurred, after the initial shock parents could only try to limit further damages by making lifestyle changes, particularly as regards the consumption of water and local food, trying to reduce the bioaccumulation of PFAS in the body and thus find some form of adaptation to a polluted environment.

As revealed by the interviews, the psychological impact derived from the empirical analysis as a construction of themes (and subthemes) is a process that, from the initial shock, becomes an awareness of a forced and necessary coexistence with PFAS by means of efforts to reorganise the family’s lifestyle. It is a psychological passage characterised by two transitions: loss of innocence and environmental adaptation. The phenomenon of transition has a long history, and it is conceptualised in the scientific literature in several ways [82]. In our study, it indicates a ‘transformation’ that forces parents to reorganise their own existence and role to cope with and adapt to new circumstances. For the parents interviewed, the transition occurred because of a rupture in their life context [83], causing a forced change that resulted in the need to build a new social and family reality. The two transitions that emerged from the interviews occurred because the participants acquired knowledge of the changes that were happening. This awareness was followed by a commitment on the part of the parents to immerse themselves in the transition process and engage in activities such as seeking information or support, recalibrating educational efforts, identifying new lifestyles, and modifying previous routines. Therefore, it is the level of awareness that influences their involvement in the management of the psychological impact caused by PFAS pollution. In fact, thanks to this acknowledgment, it is possible to reinterpret what is happening and strive to try to reorganise their parental roles and family organisation [84]. As reported in a recent review [44], chronic environmental contamination (CEC) is the experience of living in an area where hazardous substances are known or perceived to persist over time in air, water, or soil at elevated levels. Certainly, it poses toxicological health risks for directly exposed people, but the experience of living in an area exposed to long-term environmental pollution can also be psychologically stressful for the members of an affected community [85,86,87].

Based on our study of PFAS-contaminated areas, we believe that one source of this stress may be the tensions suffered in the parental role. For this reason, in the context of CEC phenomena, we suggest introducing the notion of chronic role strain (CRS) to explain a continuous tension that parents have to endure to meet the responsibility connected to their parental role and reorganise and lead their family life in the daily presence of the ‘PFAS enemy’.

Overall, the three identified phases (shock, change in lifestyle, and living with PFAS) are all characterised by uncertainty and health concerns. The literature on role strain shows how these tensions generate stress as they threaten a sense of mastery and self-esteem [48].

In the specific case study investigated here, parents reported that they felt violated and deprived of the skills (including information) to deal with the situation, thus feeling a sense of helplessness. In addition, the parental role was overloaded with thoughts of behavioural self-blame for failing to protect children against exposure to pollutants in the past. Moreover, mothers lived, and live, with the inner anguish of their awareness that they had passed on PFAS to their children during pregnancy and breastfeeding.

The very fact of being forced to live with PFAS means that these role strains become a relatively stable presence in the life of these parents, increasing their power to interfere with the way parents think about themselves and their own roles. The greater presence of mothers than fathers among the participants in this study on the one hand confirms the higher concern among the female gender generally encountered in the literature concerning environmental pollution [88,89] and, on the other, points out that those who, due to gender and cultural characteristics, are invested with the social role of caring for offspring are more affected by the psychological impact of environmental pollution, perceiving a strong parental tension and consequently taking action. An indirect confirmation of this tension of the parental role is also given by the high levels of agreement among parents when the Veneto Region first called on them to participate in the health surveillance program to monitor the presence of PFAS in the body [90].

Of course, experiencing CEC and feeling the weight of a CRS motivate people to commit themselves to try somehow to correct and/or limit the circumstances behind such threats. In our study, parents claimed to be committed to building a support network. Indeed, social resources are important to mitigate role strains. The most common social resource is social support, which may include listening, sharing, encouragement, and transmission of information [91]. The main way they tried to cope with CRS was by building a network of parents to study PFAS pollution, circulate information in the community, organise claims and protest actions, and constitute reference points (e.g., websites and social networks) to offer support to other parents and to the entire population.

The present study is not without limitations. For example, research participants are likely to belong to that category of people who, in the presence of serious environmental pollution, become aware of what has happened and do not deny/ignore the situation [92]. Additionally, they are parents in a strong social support network built to address PFAS contamination. Therefore, the PFAS affair is the central topic of their social exchanges. The interactions that occur in this group context, in comparison with the general population of the areas of Veneto Region contaminated by PFAS, might contribute to amplifying the risk perception. Consequently, these parents are more motivated to change their lifestyles and adopt measures to protect their health. Thus, it remains to be investigated whether and how we can talk about CRS for those parents who do not explicitly recognise the problem of environmental pollution. In addition, it should be considered that the participants in this study belong to the middle class of the Venetian population, a population segment with good social and cultural capital. Therefore, the study did not include fathers and mothers belonging to more marginal strata, such as low-income households or ethnic minority families. Future studies should use culturally appropriate approaches to reach marginalised communities and explore their experiences of exposure to contamination. Moreover, as role strain can also be generated by having to assume several roles at once [48], it remains to be investigated how a parent who is already weakened by the burden on children caused by toxic pollution can successfully cope with the existential situation in which they also become an environmental activist. It is plausible that this second role could accentuate their CRS, as they have two entities to deal with: the family and the community. Furthermore, more analysis is needed in regard to the age of children. It is likely that the role strains are different for parents of minor children than they are for parents of adult offspring, as various factors come in to play, one being that children are perceived as more vulnerable than adults. Finally, the present study focused exclusively on the CRS of mothers and fathers exposed to CEC. Emphasizing parental figures in childcare and family protection, it did not directly explore the presence or absence of public support resources. When people face uncertainties, concerns related to health risks and the possibility of being further exposed to contaminants every day (for example, through water consumption), coping resources external to the family (for example, at school, which provides environmental education) could be a mitigating factor for CRS. The matter will have to be deepened in further studies to improve the relationship between families and communities while facing environmental contamination.

## 6. Conclusions

Our analysis of PFAS pollution in the Veneto Region (Northern Italy) is part of the wider literature on environmental pollution. The literature has demonstrated that human-made, slowly evolving environmental disasters such as mountaintop mining [93], oil spills [94], or PFAS pollution [50,54] have negative psychosocial impacts on communities [95,96] whose members have to face many stressors [97,98]. In recent qualitative studies, a pervasive uncertainty about health effects has emerged as a key characteristic related to how PFAS contributes to stress [54,99]. This is particularly true for families with children: parents’ anxiety regarding the health and future quality of life of their children, combined with their sense of responsibility towards them, are often the most troublesome stressors faced [100]. Since the psychological impact of environmental pollution is poorly considered by the political and health authorities responsible for managing such disasters, we believe that attention to the role strains experienced by parents can contribute to developing and strengthening the necessary awareness. CRS is constituted by the persistence of unwanted and harmful conditions of distress. For parents forced to experience CEC, the persistence of these strains is an indication that it is out of their power to avoid them. CRS is a daily reminder for parents of their inability to change the unwanted conditions in which they are entangled. To the extent that parents feel they are not able to control the forces that significantly affect their lives and those of their children, the adversities they face in order to live with the PFAS can negatively affect their self-image. They represent, in the first place, the inability to control the destiny of their own family or to remove those aspects of life that are particularly harmful. Secondly, insofar as the continuing role strains are interpreted as a crisis or failure of the parental role, they can induce a process of self-blaming, up to the extreme of self-denigration. The results of our study suggest that CRS might have an erosive and threatening effect on the image of oneself as a parent. As role strains directly involve the self of parents, and threats to the self directly lead to stress, role strain indirectly results in stress [48]. Although the results of our study may be relevant for environmental pollution and health in general, they are particularly relevant for PFAS pollution, a global problem which has an impact on countless communities and important implications for human rights, the protection of the environment, and the use of water.

## Figures and Tables

**Table 1 ijerph-19-14761-t001:** Timeline of PFAS contamination in the Veneto Region.

1966	The Rimar Company began to produce Per- and Poly-Fluoroalkyl Substances (PFAS) in Trissino.
1977	Local Media announced the pollution of the aquifer by benzotrifluorides. The origin is identified as the Rimar Company.
1990	Miteni knew about the contamination.
2006	The European PERFORCE project was launched.
	Efsa imposed first European safety limits of PFAS concentrations.
2009	Through private analysis, Miteni found high levels of PFAS in their workers’ blood. No communication was had with Institutions.
2011–13	Water Research Institute IRSA-CNR carried out MATTM-CNR study to assess the environmental and health risk associated with PFAS substances in the Po River Basin and in the main Italian river basins.
2013	Regional Agency for Environmental Protection and Prevention (ARPAV) identified Miteni as responsible for the pollution.
	Veneto Region started to identify the Municipalities with contaminated drinking water.
	Activated carbon filters were installed by the water-service companies in the treatment plants of the public drinking-water distribution system.
2014	Superior Institute of Health (ISS) indicated PFAS performance levels.
	Veneto Region began food monitoring.
2015	Environmental Health Perspective published the Madrid Declaration about PFAS.
	Veneto Region and ISS applied an explorative biomonitoring study on exposed and unexposed residents and farmers.
2016	ARPAV identified 17.164 ng/l of PFAS in the discharge treatment in Trissino plant.
2017	Veneto Region established an Inquiry Commission for polluted water in Veneto related to PFAS contamination.
	The Health Surveillance plan on exposed residents in Red Area started.
	The Health Surveillance plan was extended to ex-Miteni workers and exposed people.
	Veneto Region imposed new PFAS limits.
2019	The Community Health Resilience (CHR) project at the University of Padova was launched.
2020	The CHR first study: research data were collected from a group of resident parents.
2021	The trial against Miteni Company started in Vicenza.
2021	Visit by the United Nations Special Rapporteur on toxics and human rights, Marcos A. Orellana.

**Table 2 ijerph-19-14761-t002:** Characteristics of the Study Participants (*n* = 32).

Gender	Female	25
	Male	7
Age	Mean	52.63
	Range	39–65
	SD	7.53
Family status	Married	27
	Living with a common-law partner	3
	Separated/divorced	2
Children	Under 18	31
	Mean	12.52
	Range	1–18
	SD	4.37
	Over 18	34
	Mean	26.35
	Range	19–43
	SD	5.74
Education level *	Higher education	7
	Tertiary education	23
	Secondary education	2
Occupation	Company employee	13
	Entrepreneur/freelance	7
	Teacher/educator	4
	Nurse	2
	Retired	4
	Housewife	2
Years of residence	Mean	35.86
	Range	10–62
Homeowner	Apartment	1
	Home with garden and land	31

* Higher: university/doctoral level; tertiary: middle education; secondary: lower.

**Table 3 ijerph-19-14761-t003:** The Process of Psychological Impact of the PFAS Contamination in Parents.

Phase 1		Phase 2		Phase 3
**Theme**		**Theme**		**Theme**
Shock around the discovery (100.00%)	**First transition**	Lifestyle change (100.00%)	**Second transition**	Living with PFAS (100.00%)
**Subthemes**		**Subthemes**		**Subthemes**
Unanticipated event (100.00%)		Stressful role demands (100.00%)		Invisible enemy (75.00%)
Ignorance and information needs (93.75%)	Loss of innocence	Forced protective behaviours (90.62%)	Environmental adaptation	Parental guilt (78.12%)
Personal violation (87.50%)				Social support (96.87%)
				Civic engagement (81.25%)
Uncertainty and Health concerns (100.00%)				

Note. Percentages of participants that endorsed each theme and subtheme are indicated in parentheses.

## Data Availability

The data presented in this study are available upon request from the corresponding author.

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
