# Peer review of "The Psychological Impact of Per- and Poly-Fluoroalkyl Substances (PFAS) Pollution in the Veneto Region, Italy: A Qualitative Study with Parents"

_ijerph, 2022, doi:10.3390/ijerph192214761_

Round 1

Reviewer 1 Report

The article is excellent from all points of view. The topic of psycho-social reactions in contaminated communities is very important, especially for a case of “slow emergency” like this. In Europe, unlike the United States, the academic literature on specific PFAAS pollution is not yet widespread.

The only clarification that is suggested concerns the research design. In general, the research project using CHR is effective, and it is adequately clarified. A doubt concerns the exclusive connection with a single grassroots movement (MammeNoPfas) which is not the only movement active in Veneto on the PFAAS crisis.

This selection is appropriate, but it restricted the group of 32 interviewees to the typical sociographic characteristics of the middle class (income, education level, age, housing style, etc.).

Authors specify these aspects and take them into account, but the link between the characteristics of the “sample” and some results could be made clearer, maybe in the Discussion and Conclusion.

For example, being a grassroots movement centered on the role of “mothers”, with a vision of motherhood socially built around care, this can influence the results with respect to oversizing family management among the consequences of the discovery of contamination. Second: asking for more information and knowing how to read public data on pollution, or the results of biomonitoring, requires social skills possessed only by middle class (i.e., cultural or human capital, and social capital in the engaging in a trial against managers).

These are only marginal comments, which the Authors may not even take into account in rereading the essay, or they can use a paragraph of the Conclusions to express some caution about the extensibility of the search results.

Reviewer 2 Report

This is a well-written summary of timely and important research. I have only two comments.

1. Please provide supporting evidence of the severe toxicity of PFAS that was known in the early 2000s (line 40). The provided references indicate the presence of PFAS in wildlife, but not toxicity.

2. Please provide a table of the descriptors used in the data analysis and the percent of participants who endorsed each theme. As much of the discussion is anecdotal, such a table would provide greater context and substance to the conclusions.

Reviewer 3 Report

Marialuisa Menegatto et al. submitted to IJERPH an article focusing to a qualitative study on the psychological impact of PFAS pollution in Veneto Region. The article appears well structured, with a rich and suitable bibliography and it represents a topical issue for the Regional Public Health and beyond. The broad overview of the problem dealt with and the description of the state of the art are appropriately and in detail described.

Here are my suggestions for improvement:

LL 166-168: if necessary, please specify that the biomonitoring study was conducted with the support of the Prevention Departments of Local Health Authorities (AULSS) and with specifically appointed personnel, including Environmental Health Officers and Health Visitors (if useful , please consider the following valuable citation suggestion: https://doi.org/10.3390/healthcare10101906), also to support activities in favor of the general population and specifically enlisted agricultural and agro-zootechnical workers.

LL 221-223: 32 community members were recruited, but it would also be better to specify the total number of eligible members (denominator). Have you enrolled members belonging to the "Mamme no PFAS" movement, it would probably be appropriate to consider that these members perceive the problem of exposure to PFAS in an increased way compared to the general population: this aspect could have led to biases or to possibly potentially different responses compared to a common respondent subject? If I correctly understand, is this what you mean when describing the "limitations of the study" in LL 751-753?

The English language is easily understood and the citations are appropriate to the work described.

Thank you for your efforts in perfecting this important manuscript.
